# Powder Plasma Transferred Arc Welding of Ni-Si-B+60 wt%WC and Ni-Cr-Si-B+45 wt%WC for Surface Cladding of Structural Steel

**DOI:** 10.3390/ma15144956

**Published:** 2022-07-16

**Authors:** Augustine Nana Sekyi Appiah, Oktawian Bialas, Artur Czupryński, Marcin Adamiak

**Affiliations:** 1Materials Research Laboratory, Faculty of Mechanical Engineering, Silesian University of Technology, 18A Konarskiego Street, 44-100 Gliwice, Poland; augustine.appiah@polsl.pl; 2Department of Engineering Materials and Biomaterials, Faculty of Mechanical Engineering, Silesian University of Technology, 18A Konarskiego Street, 44-100 Gliwice, Poland; oktawian.bialas@polsl.pl; 3Welding Department, Faculty of Mechanical Engineering, Silesian University of Technology, 18A Konarskiego Street, 44-100 Gliwice, Poland; artur.czuprynski@polsl.pl

**Keywords:** PPTAW, microstructure, abrasive wear, surface layer, SEM, precipitation, hardness

## Abstract

Increasing demand for sustainable approaches to mining and raw material extraction, has prompted the need to explore advanced methods of surface modification for structural steels used in the extractive industry. The technology of powder plasma transferred arc welding (PPTAW), was used in this study as a surface modification technique to improve upon the abrasive wear resistance of structural steel grade EN S355. PPTAW process parameters, namely, plasma transferred arc (PTA) current and plasma gas flow rate (PGFR), were varied, and the effects of the variation were studied and used as criteria for selecting optimum conditions for further studies and parametric reproducibility. Two metal matrix composite (MMC) powders were used in the process, having compositions of Ni-Si-B+60 wt%WC (PG) and Ni-Cr-Si-B+45 wt%WC (PE). Microstructural observation under a scanning electron microscope (SEM) revealed a dendritic, multi-directional microstructure consisting of partially dissolved primary tungsten carbide particles and secondary tungsten carbide precipitates within the MMC solid solution. The hardness of the surface layers was higher than that of a reference AR400 steel by more than 263 HV. Final surface layers obtained from the MMC powders had abrasive wear resistance up to 5.7 times that of abrasion-resistant reference AR400 steel. Alloying the MMC matrix with chromium increased the hardness by 29.4%. Under the same process conditions, MMC powder with 60 wt% WC reinforcement had better abrasive wear resistance by up to 45.8% more than the MMC powder with 45 wt% WC.

## 1. Introduction

During the service lives of engineering materials, there is constant exposure to conditions, such as creep, wear, corrosion, etc., that have significant damaging impacts on the properties of the materials. With time, there is material failure, mostly accompanied by catastrophic consequences, workplace hazards, financial loss, inter alia. The use of structural steel is frequently seen in processes such as underground mining, or oil and gas extraction. These processes subject the steels to rapid wear and failure. A major contribution to this observation in the extractive industry is the complex nature of the layered rocks as well as the geological inhomogeneity of these structures in the extraction sites [1]. The surfaces of the tools and machines used for drilling activities in this sector form the basis for accelerated material wear. The surface properties are, therefore, of much influence on the overall performance of the material. In the extractive sector, handling and transporting raw materials present an inevitability of contact of tool parts with hard components, including sand and rocks. Whereas industries in the heavy machinery and power generation sector constantly develop novel approaches to addressing challenges of material wear, the fossil fuel sector is still plagued by this limitation, due to the complexity of operation and interaction of tools with different particles [2]. The major approach adopted by such industries to address this is keeping an inventory of spare parts of machines and tools that are frequently exposed to wear [3]. This, however, encourages the need for regular downtimes for maintenance, resulting in massive tolls taken on company finances and time management, originating from the cost of tool parts, which keeps increasing, as well as both scheduled and abrupt maintenance times. Another approach adopted by the mining industry is the purchase of tools and machine parts with high quality, that can be in service for longer periods, reducing the need for regular downtimes [4]. The downside to this strategy is the high cost of such tools and machine parts, which would also need to be replaced at some point anyway. The conditions to which tools are exposed vary geologically, making it difficult to predict the service lives of the same tool used in different locations. The type of environment and abrasive wear encountered by the tools, therefore, have direct effects on their service lives.

This challenge in the extractive industry has been highlighted by research [5,6,7,8], stressing the need for reducing the abrasive wear of tools and machine parts. As some researchers [9,10,11] focus on designing new tools, by optimizing parameters to extend the service lives of such tools, surface modification approaches have been acclaimed as being more sustainable in addressing abrasive wear challenges in the extractive sector. By using surface modification strategies, techniques and materials, engineering materials, structural steel inclusive, have experienced enhanced performance and properties, such as corrosion resistance, hardness, wear resistance, etc. Surface modification, due to its flexibility of use and wide availability of material choices and techniques, is considered an economical and highly effective way of addressing challenges of engineering materials failure. Surface modification can completely alter the structure of the material’s surface, giving rise to different material properties, evidenced by the structure—property relationship of materials. Amongst the several surface modification techniques currently in use, such as physical vapor deposition (PVD), chemical vapor deposition (CVD), thermal spray, etc., the most effective strategies that are used for wear resistance applications are those involving laser and plasma technologies, which are high energy density processes [12]. Plasma transferred arc welding (PTAW) has been used for such applications, because it has advantages such as better adhesion and relatively lower dilution, compared to other techniques. The earliest use of the PTAW technique was reported in the early 1960s, for the production of overlays [13]. The PTAW techniques use a torch cathode, inside of which lies an electrode, e.g., tungsten, and a plasma arc is built around this electrode. The anode is the substrate material onto the surface of which the coating material, e.g., powders, are deposited, with shielding from an inert gas. The shield gas is used to limit oxidation during the melting of the coating material. This technique makes it less challenging to achieve desired conditions regarding dilution and penetration, enabled by the high ionization of the PTAW process [14]. PTAW can be considered a sustainable method for reasons such as low cost of implementation, high quality final surface layers, better adhesion between the substrate and surface layers, highly stable energy flux and high efficiency of melting and deposition [15]. Depending on the type of filler material, the process is called either plasma arc welding (PAW) or powder plasma transferred arc welding (PPTAW), when the filler material is a wire or powder, respectively. PPTAW generally proceeds with lower melting heat demands. The resulting property of the final product has contributions from the type of powders used, and the PPTAW process parameters, such as plasma gas flow rate (PGFR) and plasma arc current.

The flexibility of PPTAW to be used with different powders has made it a popular new surface modification technique for enhanced strength, hardness, corrosion, and abrasive wear resistance applications. Table 1 presents some metal matrix composite (MMC) powders used for PPTAW surface modification, as reported by other studies. The use of MMC with carbide reinforcements, such as WC, TiC, etc., fosters the combination of the properties of the metal matrix, such as plasticity, corrosion resistance and wear resistance, with that of the hard carbide reinforcement [16]. In application, such surface layers enable the material to withstand high impact loads and to perform better against abrasive wear. These properties are relatively hard to achieve when the surface layer is made up of only metal or only ceramic powders. This has motivated the use of MMC powders for the surface layers to address abrasive wear resistance challenges in tools and machine parts in the extractive industry. An advantage MMCs have over monolithic alloys is that the MMC could transfer higher tensile and compressive stresses than the monolithic alloy. This is enabled by the high degree of dispersion of the carbide phases in the matrix, where the existential bonds between them help transfer applied load from the matrix phase to the reinforcement [17].

Similar studies in this area of research [2,18,19] have focused on the mechanical properties and wear behavior of PPTAW hard-facing of structural steel with MMC powders of different compositions. The microstructural behavior of the applied surface layers has direct translation into the resulting properties of the layers, and it is, therefore, critical to study how the microstructure is modified as the process conditions are varied. To the best of our knowledge, at the time of conducting this study, there have not been results reported from the literature which focus on the effects of the PPTAW process parameters, and the use of different alloying elements in Ni-based MMC powders reinforced with ceramic WC particles, on the microstructure, hardness, and wear properties of the obtained surface layers on a structural steel substrate for applications in the extractive industry. Findings from this research will therefore contribute to providing more insight into this state.

**Table 1 materials-15-04956-t001:** Chemical Composition of MMC powders used in PPTAW processes for abrasive wear resistance applications, as reported from literature.

Substrate Material	Metal Matrix Composite (MMC) Powders		References
	Matrix	Reinforcement	
Aluminum	Al-Ni	SiC, TiC	[17]
Stainless steel	Co-Mo-Si-Cr-Ni-Fe	WC-Cr, Al_2_O_3_	[18]
Stainless steel	Co-Cr-W-C	WC, TiC, NbC	[19]
Titanium	Ti	NbC	[20]
Structural steel	Co-Cr-W-C	Cr_3_C_2_	[21]
Structural steel	Co-Cr-W-C	W-TiC+PD	[2]
Structural steel	Fe-C-B-Mn-Si	B_4_C	[22]
Structural steel	Fe-Cr-C-Ni	Cr_3_C_2_, Cr_7_C_3_, Cr_23_C_6_	[23]
Structural steel	Ni-Cr-B-Si	WC/Co	[24]

## 2. Materials and Methods

### 2.1. Sample Preparation

The substrate material used in this study was structural steel grade EN S355. Steel plate specimens were prepared according to the specifications indicated in Figure 1a. The powder plasma transferred arc welding (PPTAW) technique was used to deposit the MMC powders onto the surface of the substrate material. The process was carried out using PPTAW system EuTronic Gap 3511 DC synergic (Castolin Eutectic, Gliwice, Poland), shown in Figure 2. The cathode was made up of a plasma torch consisting of a central tungsten electrode around which a plasma arc was built. The powders were fed through the torch. The anode was the structural steel plate substrate material. Argon gas was used as the shield to protect the padding weld from possible oxidation resulting from interaction with the environment during melting. To obtain the optimal process parameters for the PPTAW surface modification technology, process parameters were varied. The parameters varied were the plasma gas flow rate (PGFR) and the plasma transferred arc (PTA) current. Varied values of the process parameters are given in Table 2. Varying the process parameters provided additional control over the technique for easier reproducibility. Optimum process parameters are those that proceed with high deposition rate, low dilution, less distortion, and the least surface imperfections. In Figure 3, representative images of the specimen after the PPTAW process are presented for each powder used. Under the same process conditions, the final surface clad obtained from the various powders were not significantly distinguishable from each other. Ni-based MMC powders were obtained commercially from Castolin Eutectic^®^ in Gliwice Poland. Two types of MMC powders were acquired with varying concentrations of WC reinforcement. Powder designated by PG has chemical composition Ni-Si-B+60 wt%WC and powder designated by PE has chemical composition Ni-Cr-Si-B+45 wt%WC. The concentration of WC was varied for both powders to study the effects of the reinforcement concentration on the mechanical properties of the final surface clad. This would also help in understanding the mechanism of abrasive wear better. SEM morphological images of the powders were generated using Supra 35 (Zeiss, Oberkochen, Germany). MMC powder particle sizes were analyzed using laser particle sizer ANALYSETTE 22 (Fritsch, East Windsor, NJ, USA).

### 2.2. Characterizaion and Testing Methods

#### 2.2.1. Microscopic Observations

The digital microscope, Leica DVM6 (Leica Microsystems, Heerbrugg, Switzerland) was used to obtain digital images of the surface layers, and to examine the depth of wear on the coated surfaces after the abrasive wear resistance tests. Light microscopy was used to generate micrographs of the deposited MMC after the PPTAW process. The light microscope used was AxioVision (ZEISS, Jena, Germany).

#### 2.2.2. Hardness Tests

The hardness of the as-deposited surface layers was measured using the Rockwell hardness testing equipment SHR-15E (GSTI Co. Ltd., Guiyang, China), and a load of 150 kg. The test was carried out on the extreme surface of each tested sample after thorough surface cleaning with compressed air. In total, 8 measurements were taken at an even space of 10 mm between the points. The average hardness was then computed and recorded. The hardness tester FM-ARS 9000 (Future Tech Corporation, Tokyo, Japan) and a load of 9.81 N was used to measure the Vickers hardness of the surface layers. These microhardness measurements were carried out along the cross-section of the clad after metallographic polishing, with 8 measurements taken at random points in the matrix as well as the reinforcing WC. The average was then computed and recorded.

### 2.3. Relative Abrasive Wear Resistance Measurement

#### 2.3.1. Abrasive Wear Test

Abrasive wear resistance tests were performed on the as-deposited MMC surface layers as well as on a reference material (abrasive wear resistant AR400 steel). The test followed the guidelines of standard ASTM G65 which is often referred to as the “rubber wheel” method. This method sandwiches abrasive materials between the surface under study and a rubber lined wheel, as shown in Figure 4. The rubber lined wheel is set to revolving motion while the surface under study is set in a stationary position over a period. The process parameters used in this test are outlined in Table 3.

#### 2.3.2. Calculation of Relative Abrasive Wear Resistance

To determine the abrasive wear resistance of the prepared surface layers relative to that of the reference material, firstly, the weights of the tested specimen were taken before and after the test. The mass loss was then determined using Equation (1). The volume loss was estimated using Equation (2). Then, the relative abrasive wear resistance of the specimen under study was computed with Equation (3).
Mass Loss, M_L_ [g] = M_B_ [g] − M_A_ [g](1)
(2)$$\mathrm{Volume}\;\mathrm{Loss},\;V_{L}\;[{\mathrm{mm}}^{3}]=\frac{M_{L}[g]}{\rho [\frac{g}{cm^{3}}]}\times 1000$$
(3)$$\mathrm{Relative}\;\mathrm{Abrasive}\;\mathrm{Wear}\;\mathrm{Resistance}=\frac{V_{\mathrm{LR}}[mm^{3}]}{V_{\mathrm{LS}}[mm^{3}]}$$
where M_B_ is the mass of the specimen before abrasive wear test; M_A_ is the mass of the specimen after abrasive wear test; ρ is the density of the material; V_LR_ is the average volume loss of the reference material; V_LS_ is the average volume loss of the specimen.

## 3. Results and Discussion

### 3.1. MMC Powder Morphology and Particle Size Distribution

SEM images of the powders PG and PE are shown in Figure 5a and Figure 6a, respectively. The morphology of the MMC powders under the SEM revealed two distinct structures. These structures were subjected to microanalysis (Figure 5 and Figure 6), coupled with the chemical compositions in Table 4 and Table 5, revealed that the matrix particles appeared to be more spherical, whereas the reinforcing carbides had sharp-edged morphology. Spherical powder particles are more desirable in applications where high toughness is required, and sharp-edged or angular shaped powder particles are more desirable in applications where wear resistance is required. Nonetheless, MMC powder particles having a mixture of both spherical and angular morphologies results in more mechanical stability, suitable for applications requiring high toughness, wear resistance and high hardness [25]. Figure 7 shows the results of laser-assisted particle size analysis. Statistical deductions from the plots show that for both powders, analyzed with a specific quantity of particles, denoted by *n*, 10% of these particles were smaller than *d*_10_, 50% of these particles were smaller than *d*_50_, and 90% of these particles were smaller than *d*_90_. These values could be used to determine the degree of uniformity, or span of the powder particles, a dimensionless value given by the equation
(4)$$Span=\frac{d_{90}-d_{10}}{d_{50}}$$

When span is equal to 0, it signifies complete particle uniformity, and a higher value of span conversely shows a higher degree of particle non-uniformity. The PG powder had a span of 1.3 and, as shown in Figure 7b, the size distribution presented as a group, even though there were two distinct morphologies. The PE powder had a higher span value of 4.3 and, as shown in Figure 7a, the size distribution presented as two different groups, to make up for the higher disparity in the particle sizes. The PG powder particles can, therefore, be said to be relatively larger than the PE powder particles. This also accounts for the observation of larger particle sizes of PG powders (Figure 5a) than the PE powders (Figure 6a) at the same magnification under the scanning electron microscope.

### 3.2. Metallographic Analysis and Precipitation

Microstructural observation of the coated layer revealed the degree of dispersion of the WC reinforcement and its precipitates in the Ni-based matrix throughout the coating. The observable zones along the cross-section of the surface layer are shown in Figure 8.

An exemplary observation of the middle zone of sample 4-PE, prepared under conditions of 110 A PTA current and 1.5 L/min PGFR, under the SEM, is shown in Figure 9. Figure 9a shows partially dissolved primary carbides and precipitates of secondary carbides throughout the matrix. They formed a dendritic, multi-directional microstructure. The chemical composition of the area of the middle zone shown in Figure 9a was analyzed and the scattered X-ray energy diagram in Figure 9b shows the chemical elements present in this zone. The elemental maps in Figure 9c–j show the presence and surface distribution of the chemical elements present in the middle zone. The bright grey regions in Figure 9a are the primary carbides and precipitates of secondary carbide particles, evidenced by the high concentrations of carbon and tungsten in Figure 9d,i. The black regions in Figure 9a are the Ni-based matrix sites, evidenced by Figure 9i. Ceramic reinforcement particles in MMCs usually agglomerate when solidified, due to the difference in density of the carbides from the matrix [26]. The PPTAW technique has been reported to result in surface clads that have agglomerated ceramic reinforcements, weak wettability, and imperfections on the surface of the clads [27]. In this work, however, the ceramic carbides, owing to the effectiveness of the method of mixing the powder particles, were seen to be dispersed in the matrix without agglomeration. The dendritic structure of the surface layer, formed by the precipitation of complex secondary carbides on the boundary of the matrix and carbide was made possible by the partial melting of the WC particles (T_m_ = 2785 °C) and the complete melting of the Ni-based matrix powders (T_m_ = 1555 °C) during the deposition [28]. Poloczek et al. (2019) [29] have stipulated that the stability of the deposited clad is possible by the bonding of the primary carbides to the matrix by the secondary carbides through accelerated diffusion. This SEM microstructural observation was like all samples prepared by each of the PG and PE MMC powders used. The precipitates of the secondary carbides in the matrix were subjected to microanalysis and the results of this microanalysis are presented in Figure 10. It was observed from this analysis that these precipitates were composed of W, C, Ni, Fe and Si. The proportions of these compositions are given in Table 6. The dissolution of the substrate material into the surface coating during deposition accounted for the presence of Fe in the secondary carbide precipitates.

At the coating–substrate interface, shown in Figure 11, two major cross-sectional zones were observed, i.e., the dilution zone and the heat affected zone (HAZ). Microanalytical determination of the concentrations of chemical elements across the interface, in the direction from the coating to the substrate is presented in Figure 11. The concentrations of W and Ni were seen to be higher in the coating and in the dilution zone. However, they gradually dropped to their minima within the HAZ and maintained the least concentrations within the substrate material. A contrasting observation is seen regarding Fe. Its concentration was the least at the dilution zone and rose gradually within the HAZ and reached its maximum in the substrate material. Carbon maintained a constant concentration across the interface. These observations at the interface suggested a strong adhesion between the coating and the substrate material, by dissolution of chemical components across the interface from the substrate material into the coating and vice versa, this being metallurgical bonding.

### 3.3. Hardness Tests Results

Microhardness measurement results of the Ni-based matrix and WC reinforcement for surface layers prepared by each MMC powder, are presented in Table 7, in a comparative observation regarding the microhardness of the reference abrasion-resistant steel AR400. The measured average microhardness of the surface layers was observed to be significantly higher than that of the reference material by more than 263 HV. The observed high value of standard deviation was accounted for by the fact that the measurements were taken at random points throughout the clad for both the matrix and the WC phases. The non-uniform dispersion of the WC particles in the clad influenced this measurement approach. It was consequently observed that the microhardness measurements increased along the clad towards the coating–substrate interface. This could be attributed to the weld dilution at this interface, and the effects of the HAZ. Such measurement difficulties have been reported by research works [28,30], which studied the wear behavior of MMC cladded surfaces. Surface hardness of the prepared coatings are also presented in Table 8. Results from microhardness measurements on the surface were influenced by the distribution of stress from the matrix to the reinforcing WC particles. This was because the gradient of the surface of the samples comprised of MMC solid solution had dispersed primary WC particles, as well as precipitates of secondary carbides. The observed increase in hardness could be explained by the partial saturation of the matrix caused by the partial dissolution of the primary tungsten carbides in the matrix, as well as the alloying of the Ni-based matrix with Cr, in the case of samples prepared with PE MMC powder. The use of Cr as an alloying element influences the mechanical properties by increasing the hardness of the alloy [31]. Cr can form different carbides in the presence of carbon. Due to this property, it improves the hardenability of steel during production [32]. In a study conducted by Lin et al. (2018b) [31] to investigate the effects of Cr on the microstructure and properties of TiC-steel composites, it was reported that an increase in the Cr content caused an increase in the hardness, with a corresponding decrease in the transverse rapture strength (TRS). Similar results were obtained in this work: the average microhardness of the matrix of the PE MMC powder was about 29.4% higher than that of the matrix of the PG MMC powder. This was significantly attributed to the use of Cr as an alloying element in the PE MMC powder.

### 3.4. Abrasive Wear Resistance Test Results

The abrasive wear resistance of the surface layers prepared by each MMC powder, relative to the wear resistance of reference material AR400, was carried out using the methods described in Section 2.3. The results were computed using Equations (1)–(3), and presented in Table 9. The marginal mass loss for each sample set the right measurement precision to three decimal places. However, results are presented in four decimal places to make up for measurement anomalies and external environmental influences, which, in the end, did not have any significant influence on the reported relative abrasive wear resistance values. It was observed that the abrasive wear resistance of the surface layers was much higher than the reference material in the cases of both MMC powders used. It was again observed in this test that, on average, under the same process conditions, the MMC powder PG, having 60 wt% WC, had better abrasive wear performance than the PE MMC powder having 45 wt% WC, by up to 45.8%. The higher volume of WC present in the PG powder explained the observation of higher abrasive wear resistance to that of the PE powder MMC. Figure 12 shows the surface structure of the MMC surface layers prepared by each powder under the same conditions of 110 A PTA current and 1.0 L/min PGFR, after undergoing the abrasive wear test. The mechanism of abrasive wear resistance observed in this work was studied and can be described as occurring in two major stages. At the initial stage, as the rubber wheel was set in motion with the quartz sand in contact with the surface of the layer, the surface began to wear off. This continued until the quartz sand removed enough material from the surface and came into closer proximity with wear debris rich in tungsten carbide. Contact with these WC particles initially caused smearing of the WC particle. As the contact time increased, WC particles caused reduction in the friction from the quartz sand and the revolution of the rubber wheel, significantly reducing wear. The image of the surface layer after the abrasion test in Figure 13a reveals exposed WC particles at the surface when further abrasion was hindered. The depth map in Figure 13b shows that the region on the surface which lacked WC particles were abraded deeper than the regions rich in WC particles. The wear mechanism observed in the reference material, which was WC free, presented in Figure 12c and Figure 16a,e,i, shows that during the dent formation, the exposed metal surface was ploughed from the micro scale, enhancing the transfer of abrasive material onto the dent surface. This resulted in deeper grooves and displacement of metal to the sides of the abrasive material as contact time increased. This observed mechanism has also been reported in a similar study by Chung et al. (2021) [33] who studied the erosion–corrosion wear behavior of AR400 steel and found micro ploughing to be the main wear mechanism in this WC free steel. It was deduced from the studied mechanisms that MMC powders rich in WC have better abrasive wear resistance, than those with relatively lower volumes of WC. This mechanism also supported the observation of better abrasive wear resistance in samples prepared with PG powder MMC than in those prepared with PE powder MMC, which had relatively lower volume of WC reinforcement.

### 3.5. Parameter Optimization

Identifying optimum process parameters for high energy density processes like PPTAW is of much importance. This is because it significantly increases productivity by optimizing the quality and quantity of deposition [34]. PPTAW process parameters, plasma transferred arc current and plasma gas flow rate, were varied to investigate their effects on the resulting properties of the surface layers, to aid in establishing the criteria for selecting optimum values for parametric reproducibility.

#### 3.5.1. Plasma Transferred Arc (PTA) Current

The PTA current was varied between 110 A and 150 A, while keeping all other parameters constant, to prepare samples from the MMC powders. An observation of the cross-section of the surface layers revealed that increasing the PTA current caused more dissolution of the WC reinforcements into the matrix. To illustrate this, cross-sections of the surface clad of samples 2-PG and 3-PG, prepared under the same process conditions but at two different PTA current values (110 A and 150 A), are shown in Figure 14. At PTA current of 110 A, the WC particles were seen to be more evenly distributed throughout the cross-section of the coating (Figure 14a). After the PTA current was increased to 150 A, the WC particles were seen to be less clustered, having a relatively higher degree of dispersion within the matrix (Figure 14b). It was particularly observed that at higher PTA current, the subsurface of the layer in the cross-sectional view was mostly free of WC particles. This significantly reduced the abrasive wear performance of the layer, as shown in the comparative plots in Figure 15c,d.

Figure 14 also shows the measured average thickness of the coating along its cross-section. At PTA current of 110 A, the thickness of the weld bead averaged around 2.7 mm. At a higher PTA current, the average of the weld bead was around 3.3 mm. An increase in the PTA current resulted in higher overall thickness of the deposited layers. This was caused by a rise in temperature upon an increase in PTA current. This led to an increase in supplied energy, causing more dissolution of the reinforcing particles. Consequently, adhesion was also improved, due to the higher dissolution of the base material into the surface layer at the HAZ [35]. The dissolution of the WC particles in the matrix increased complex carbon compound formation with the alloying elements, such as Cr, and this slightly increased the overall hardness of the coating matrix [36]. This is also supported by the plots in Figure 15a,b.

#### 3.5.2. Plasma Gas Flow Rate (PGFR)

The mechanical effects of the PGFR on the abrasive wear resistance and hardness of the prepared surface coatings were evaluated by keeping all process conditions constant and varying the PGFR values as 1.0 L/min, 1.2 L/min and 1.5 L/min. Representative images of pre- and post-abrasive wear tests are illustrated in Figure 16, with samples 1-PE, 2-PE and 4-PE having PGFRs of 1.0, 1.2 and 1.5 L/min, respectively. As can be seen from the plots in Figure 17a,b, the relative abrasive wear resistance of the coatings, for each MMC powder used, increased from 1.0 L/min to 1.2 L/min, which could be considered as a critical value, then reduced as the PGFR was further increased to 1.5 L/min. As gas flow rate is the measure of the volume of gas passing a particular point over a period, it appears that an increase in the PGFR increased thermal activity, causing increased velocity of powder particles along the axis of the torch. This increased the compressive forces between the powder particles, and at the coating-substrate interface. As a result, there was greater adhesion between dissolved particles, reducing the overall porosity of the clad. This formed the basis for the observation of a general increase in the microhardness of the matrix of the coatings as the PGFR increased from 1.0 L/min to 1.5 L/min in Figure 17c,d, and the increase in abrasive wear resistance as the PGFR increased from 1.0 L/min to 1.2 L/min. However, as the PGFR increased, turbulent flow of the powder particles resulted in increased energy of the process, similar to increasing the PTA current, resulting in less WC particles present at the coating subsurface. From Figure 16f–h, it can be seen that at PGFR of 1.0 L/min, the grooves on the surface after contact with abrasive quartz sand were seen to be less deep, compared to the grooves on the surfaces of the samples having PGFR of 1.2 L/min and 1.5 L/min. The lower presence of the WC particles resulting from higher dissolution of the WC particles at the subsurface was the reason for the deeper grooves caused by the quartz sand at higher PGFR values and the observation of a decline in the abrasive wear resistance at a PGFR of 1.5 L/min in Figure 17a,b. These observations were consistent with other research works [37,38] that investigated the effects of gas flow rate on the mechanical and wear behavior of surface coatings using TIG and plasma spray approaches, respectively.

The optimum process parameters from those listed in Table 2, coupled with PPTAW device working parameters are summarized in Table 10. These parameters were used extensively for the study, and they formed the basis for the presented results.

## 4. Conclusions

The technology of powder plasma transferred arc welding (PPTAW) was used in this study as a surface modification technique to improve upon the abrasive wear resistance of structural steel grade EN S355 for applications in the mining and raw materials extraction industry.Two metal matrix composite (MMC) powders were used in the process, having compositions of Ni-Si-B+60 wt%WC (PG) and Ni-Cr-Si-B+45 wt%WC (PE).PPTAW process parameters, namely, plasma arc (PTA) current and plasma gas flow rate (PGFR), were varied, and the effects of the variation were studied and used as criteria for selecting optimum conditions for further studies and parametric reproducibility.PTA current had influence on the degree of dissolution of the primary carbides in the matrix, the thickness of the surface clad, the surface hardness and the abrasive resistance of the surface layer.PGFR was observed to have influence on the hardness and abrasive wear resistance of the surface coatings as it increased from 1.0 L/min to 1.2 L/min and then to 1.5 L/min.Microstructural observation revealed a dendritic, multi-directional microstructure, consisting of partially dissolved primary tungsten carbide particles and secondary tungsten carbide precipitates, within the Ni-based matrix.The hardness of the surface layers was higher than that of a reference material AR400 by more than 263 HV. Final surface layers obtained from the MMC powders had abrasive wear resistance up to 5.7 times that of abrasion-resistant reference Hardox 400 (AR400) steel. Alloying the MMC matrix with chromium increased the surface hardness by 29.4%. Under the same process conditions, MMC powder with 60 wt% WC reinforcement had better abrasive wear resistance by up to 45.8% more than the MMC powder with 45 wt% WC.

## Figures and Tables

**Figure 1 materials-15-04956-f001:**
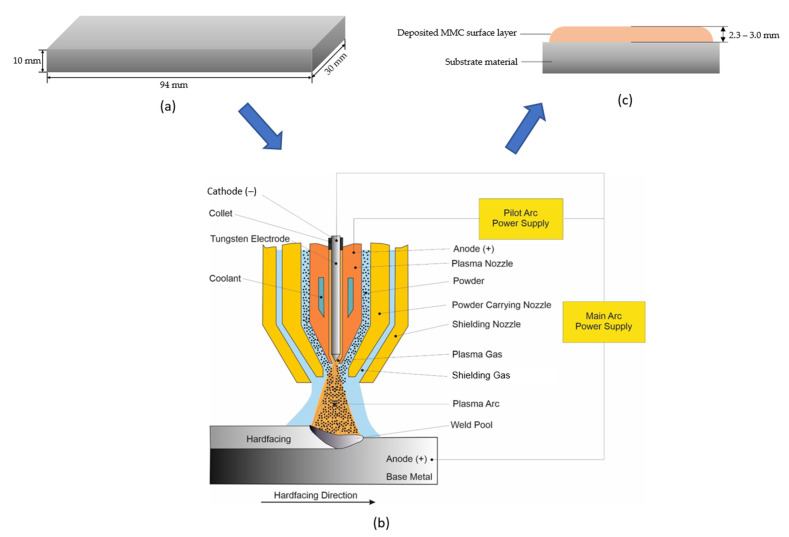
Process flow diagram for the surface cladding process (**a**) Image of the prepared structural steel plate specimen used for the study, including its dimensions (**b**) Schematic diagram of the PPTAW process (**c**) Schematic image of the cross-section of the final product after surface coating.

**Figure 2 materials-15-04956-f002:**
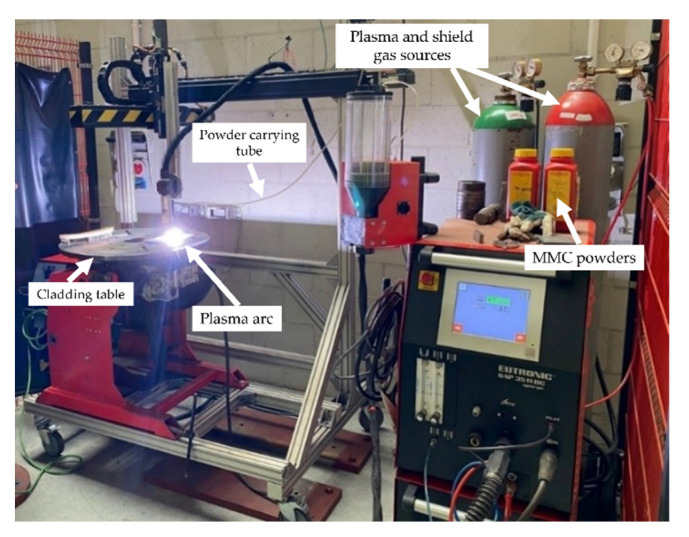
EuTronic© Gap 3511 DC synergic system used for PPTAW process.

**Figure 3 materials-15-04956-f003:**
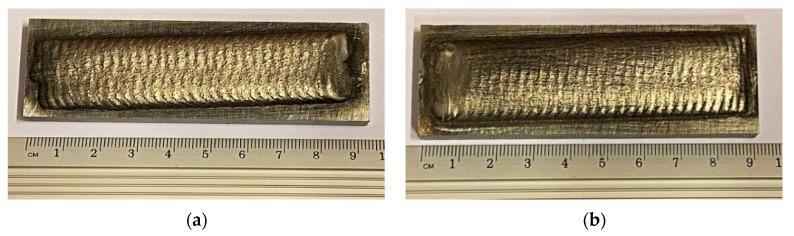
Images of prepared samples showing the as-deposited MMC surface coatings on the substrate material (**a**) specimen 2-PG prepared with PG MMC (**b**) specimen 2-PE prepared with PE MMC.

**Figure 4 materials-15-04956-f004:**
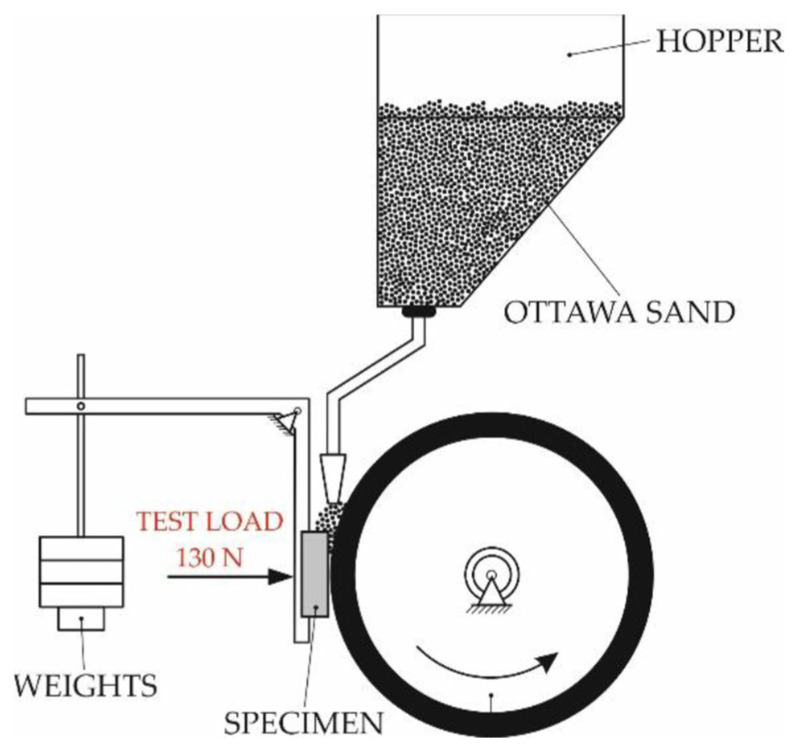
Abrasive wear resistance testing schematic diagram showing the abrasive quartz sand particles sandwiched between the rotating rubber-lined wheel and the stationary surface of the MMC hard-faced layer.

**Figure 5 materials-15-04956-f005:**
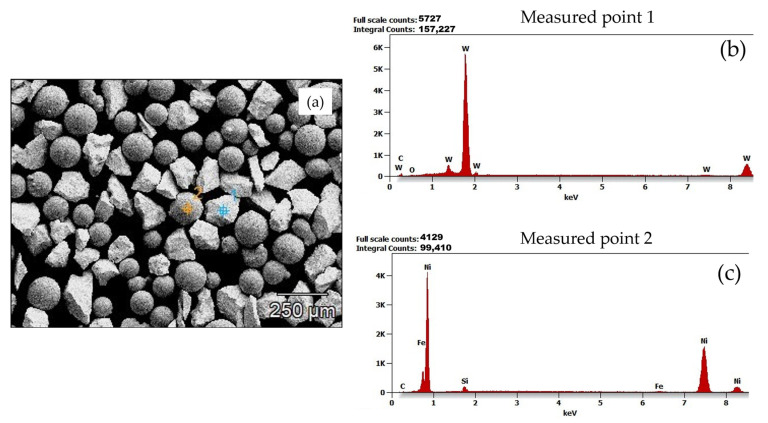
Microanalysis of powder particles (**a**) acquired SEM image of PG powder showing its morphology and measured points under analysis (**b**,**c**) Energy-dispersive X-ray spectrometry (EDS) diagrams of measured points 1 and 2.

**Figure 6 materials-15-04956-f006:**
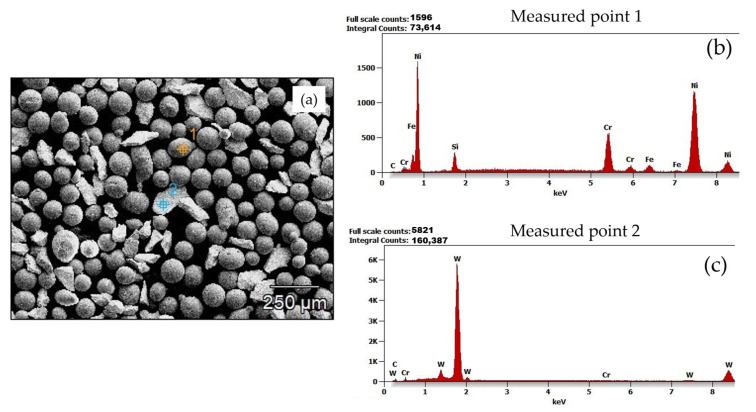
Microanalysis of powder particles (**a**) acquired SEM image of PE powder showing its morphology and measured points under analysis (**b**,**c**) Energy-dispersive X-ray spectrometry (EDS) diagrams of measured points 1 and 2.

**Figure 7 materials-15-04956-f007:**
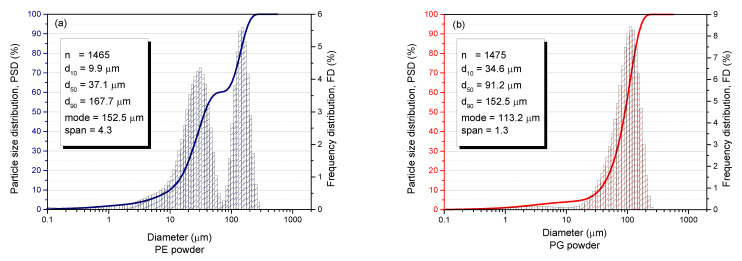
Particle size distribution (PSD) plots for the MMC powders (**a**) PE powder PSD plot with particle statistics (**b**) PG powder PSD plot with particle statistics.

**Figure 8 materials-15-04956-f008:**
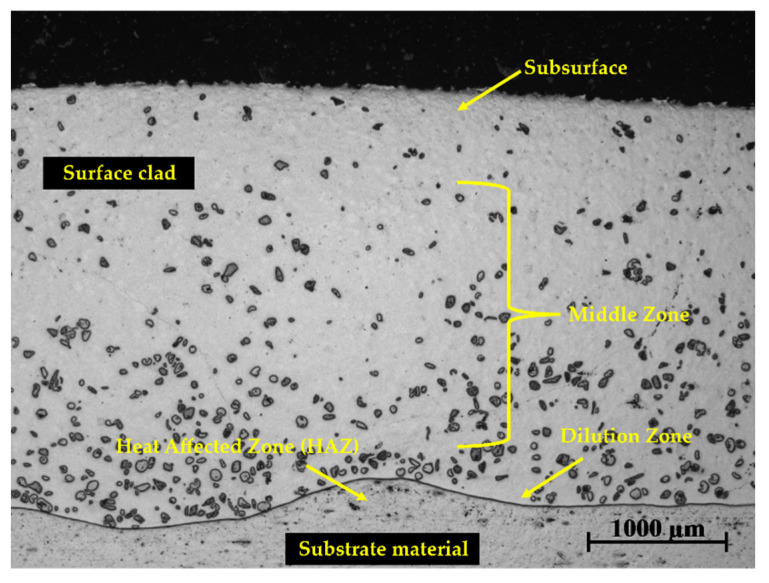
Cross-section of deposited surface layer of sample 1-PE showing the subsurface of the layer, middle zone, dilution zone, and HAZ.

**Figure 9 materials-15-04956-f009:**
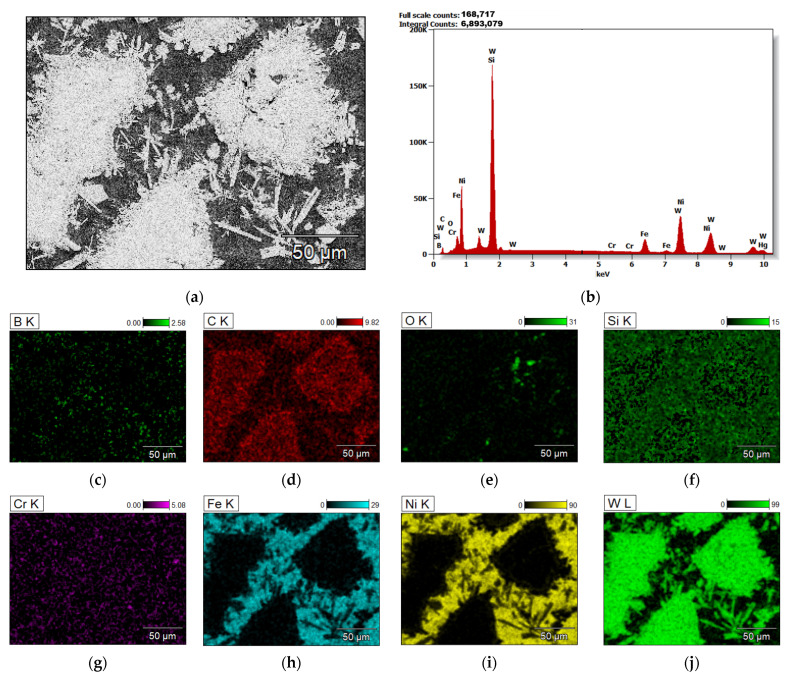
Elemental analysis of the middle zone of the surface layer of sample 4-PE (**a**) area under observation (**b**) X-radiation energy diagram of area under observation (**c**–**j**) elemental maps showing the position and amounts of elements present in the area under observation.

**Figure 10 materials-15-04956-f010:**
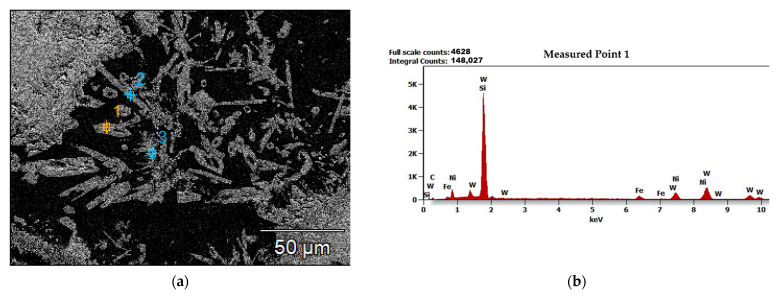
Results of microanalysis of precipitates of the WC reinforcement within the surface layer of sample 4-PE prepared under conditions of 110 A PTA current and 1.5L/min PGFR (**a**) Image of microanalytical area showing points of interests for analysis (**b**–**d**) Energy-dispersive X-ray spectrometry (EDS) diagrams of measured points 1 through 3.

**Figure 11 materials-15-04956-f011:**
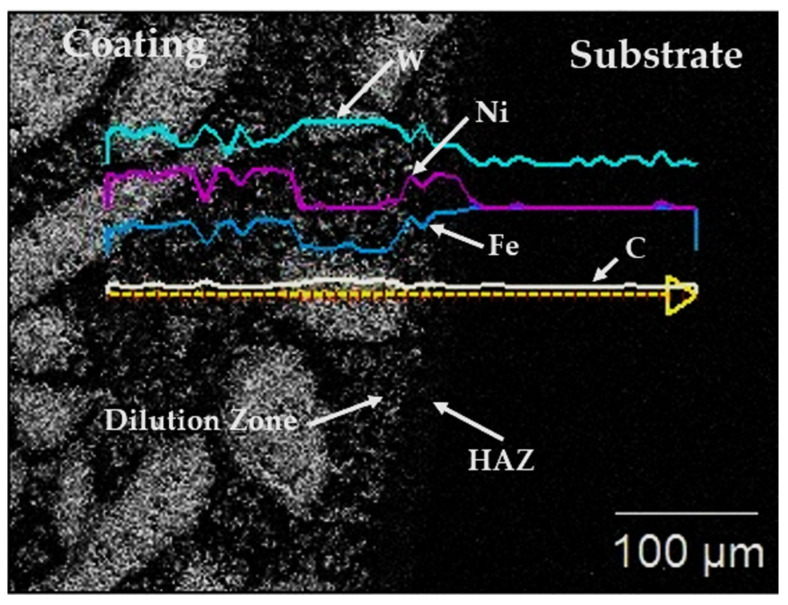
SEM image of the coating-substrate interface of sample 3-PE, showing the dilution zone, HAZ and the diffusion gradients of the chemical components involved in the interface adhesion between the coating and the surface.

**Figure 12 materials-15-04956-f012:**
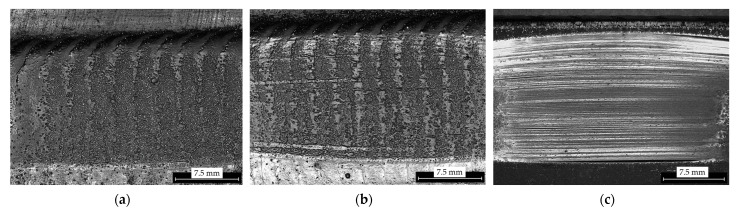
Surface of coating layers after abrasive wear resistance tests (**a**) surface of layer of sample 1-PG prepared using PG powder (**b**) surface of layer of sample 1-PE prepared using PE powder (**c**) surface of reference material AR400.

**Figure 13 materials-15-04956-f013:**
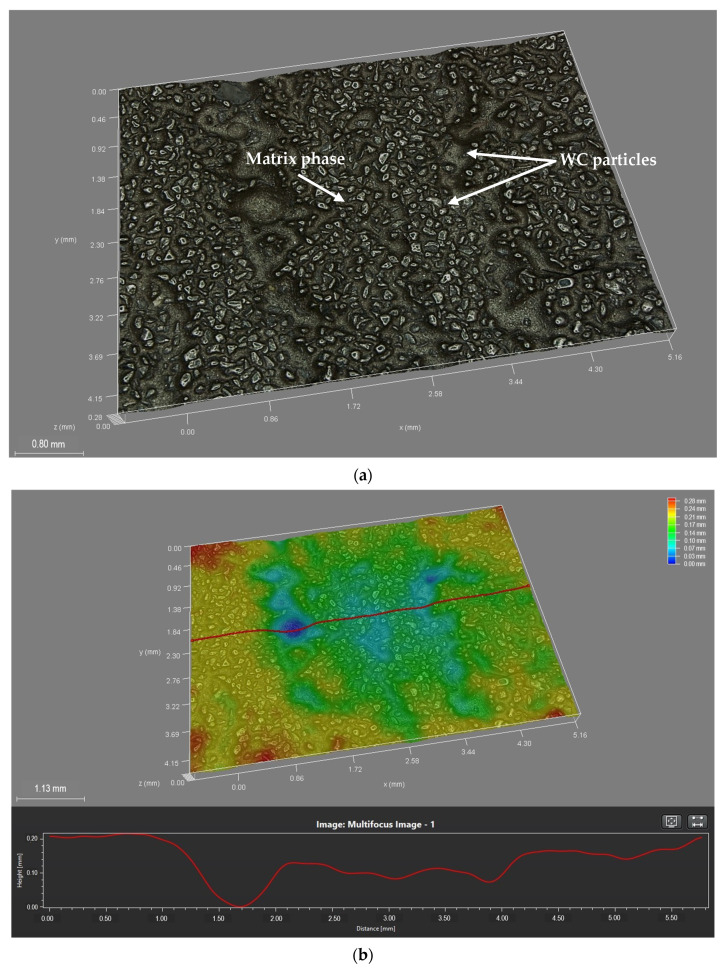
Images of surface layer of sample 4-PG after abrasive wear resistance test (**a**) post-abrasive wear test surface showing dispersed WC particles and the matrix phase (**b**) depth map showing the depth of abrasion on the surface of the layer.

**Figure 14 materials-15-04956-f014:**
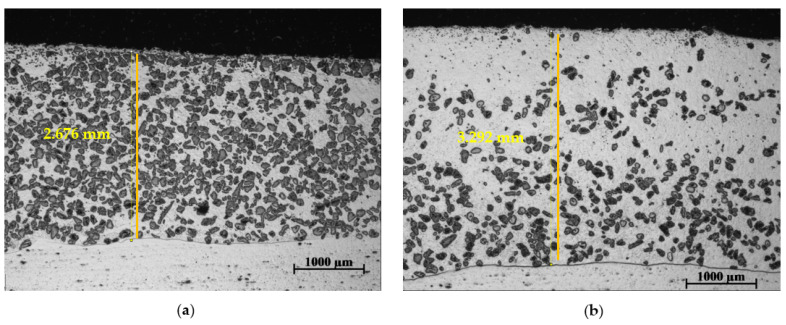
Cross-section of surface layers prepared with PG powder MMC at different PTA current values showing the measured thickness of surface layers (**a**) surface layer of 2-PG prepared at PTA current of 110 A (**b**) surface layer of 3-PG prepared at PTA current of 150 A.

**Figure 15 materials-15-04956-f015:**
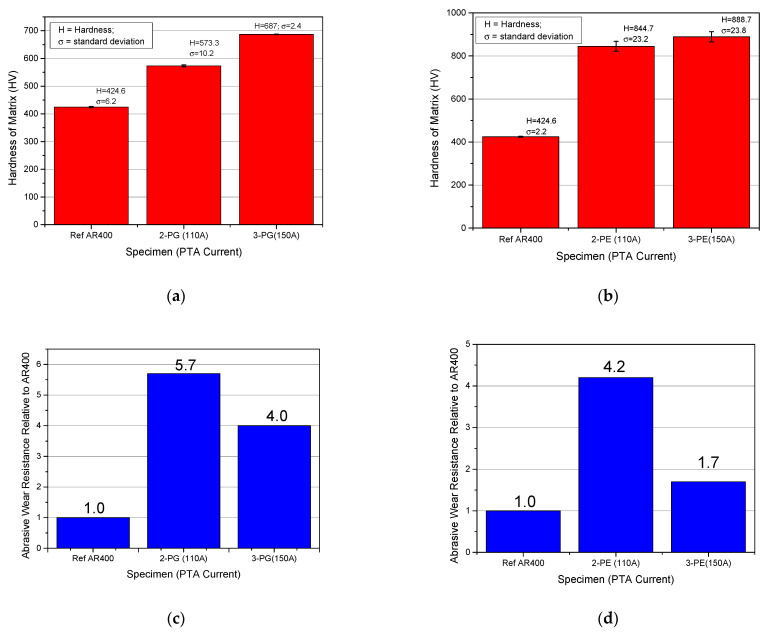
Comparative plots on the effects of PTA current on hardness and abrasive wear performance of surface coating (**a**,**b**) effects of PTA current on hardness of matrix of samples prepared with PG and PE powder MMCs respectively (**c**,**d**) effects of PTA current on the abrasive wear resistance of samples prepared with PG and PE powder MMCs respectively.

**Figure 16 materials-15-04956-f016:**
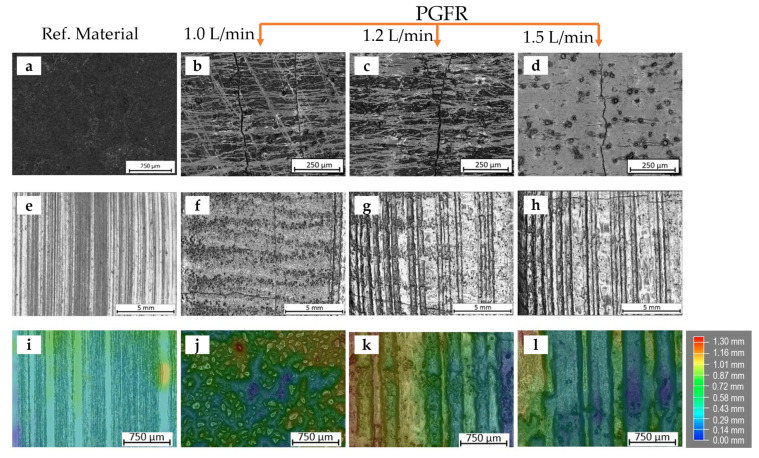
Images of the surfaces of specimens before and after abrasive wear test (**a**,**e**,**i**), reference material AR400 steel before abrasive wear test, after abrasive wear test and wear depth after abrasive wear, respectively (**b**,**f**,**j**) specimen 1-PE with PGFR of 1.0 L/min and PTA current of 110 A before abrasive wear test, after abrasive wear test and wear depth after abrasive wear, respectively (**c**,**g**,**k**) specimen 2-PE with PGFR of 1.2 L/min and PTA current of 110 A before abrasive wear test, after abrasive wear test and wear depth after abrasive wear, respectively (**d**,**h**,**l**) specimen 4-PE with PGFR of 1.5 L/min and PTA current of 110 A before abrasive wear test, after abrasive wear test and wear depth after abrasive wear, respectively.

**Figure 17 materials-15-04956-f017:**
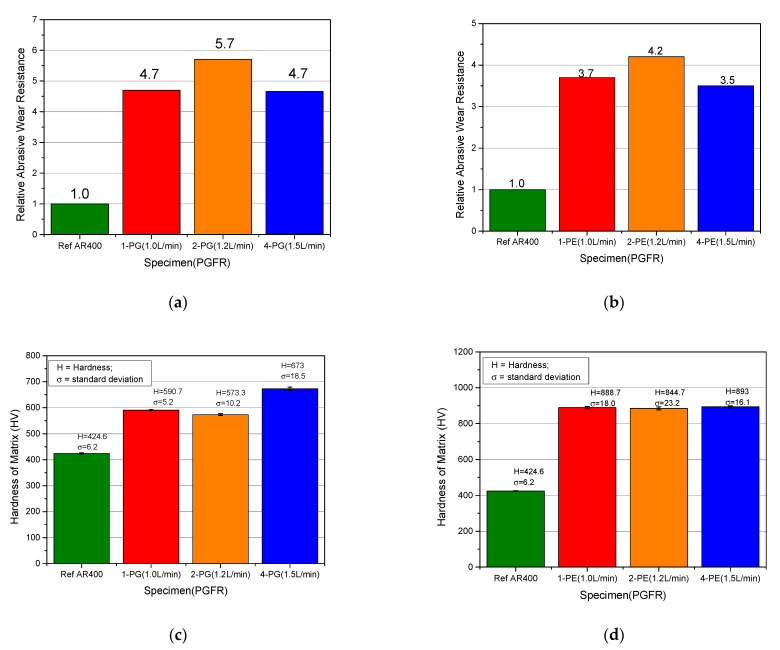
Comparative plots on the effects of PGFR on abrasive wear performance and hardness of surface coatings (**a**,**b**) effects of PGFR on the abrasive wear resistance of samples prepared with PG and PE powder MMCs, respectively (**c**,**d**) effects of PGFR on hardness of matrix of samples prepared with PG and PE powder MMCs, respectively.

**Table 2 materials-15-04956-t002:** PPTAW process parameter variation for specimen preparation, Travel Speed, V = 1.3 mm/s.

Specimen	MMC Powder	Plasma Arc Current (A)	Plasma Gas Flow Rate (L/min)
1-PG	PG *	110	1.0
2-PG	PG	110	1.2
3-PG	PG	150	1.2
4-PG	PG	110	1.5
1-PE	PE **	110	1.0
2-PE	PE	110	1.2
3-PE	PE	150	1.2
4-PE	PE	110	1.5

* PG—MMC with composition Ni-Si-B+60 wt%WC; ** PE—MMC with composition Ni-Cr-Si-B+45 wt%WC.

**Table 3 materials-15-04956-t003:** Abrasive wear resistance testing conditions.

Parameter	Value	Unit
Abrasive particle grain size	210–297	µm
Feed rate	335	g/min
Pressure	130	Pa
Rubber wheel turns	6000	turns
Test time	30	min

**Table 4 materials-15-04956-t004:** Chemical composition in weight % and atom % of measured points of PG powder under analysis.

		C	O	Si	Fe	Ni	W
Measured point 1	Weight %	3.9	0.9	-	-	-	95.2
	Atom %	36.2	6.4	-	-	-	57.5
Measured point 2	Weight %	1.5	-	2.4	0.7	95.4	-
	Atom %	6.9	-	4.7	0.6	87.8	-

**Table 5 materials-15-04956-t005:** Chemical composition in weight % and atom % of measured points of PE powder under analysis.

		C	O	Si	Cr	Fe	Ni	W
Measured point 1	Weight %	0.7	-	3.5	15.3	3.5	76.9	-
	Atom %	3.9	-	6.7	15.8	3.4	70.3	-
Measured point 2	Weight %	3.8	3.7	-	0.5	-	-	92.0
	Atom %	29.8	22.0	-	0.9	-	-	47.3

**Table 6 materials-15-04956-t006:** Chemical composition in Weight % and Atom % of measured points of precipitation subjected to microanalysis of sample 4-PE prepared under condition of 110 A PTA current and 1.5 L/min PGFR.

		C	Si	Fe	Ni	W
Measured Point 1	Weight %	3.0	5.0	3.6	10.0	78.4
	Atom %	22.6	16.5	6.0	15.7	39.2
Measured Point 2	Weight %	3.1	3.9	2.8	11.0	79.3
	Atom %	24.2	13.0	4.7	17.6	40.6
Measured Point 3	Weight %	2.6	5.1	4.8	12.4	75.1
	Atom %	19.4	16.5	7.9	19.2	37.1

**Table 7 materials-15-04956-t007:** Measured microhardness of the matrix and reinforcing carbides through the cross-section of the deposited surface layers for each MMC powder used, compared to the microhardness of abrasive wear-resistant steel, AR400.

Specimen	Microhardness of Matrix (HV1)		Microhardness of Reinforcement (HV1)	
	Mean	Standard Deviation	Mean	Standard Deviation
1-PG	590.7	5.2	2413.0	62.9
2-PG	573.3	10.2	2128.7	33.3
3-PG	687.0	2.4	2162.7	76.1
4-PG	673.0	18.5	2275.0	49.5
1-PE	888.7	18.0	2349.3	38.7
2-PE	844.7	23.2	2436.3	24.1
3-PE	888.7	23.8	2343.3	61.6
4-PE	893.0	16.1	2391.3	80.5
Reference Material				
AR400	424.6	6.2	-	-

**Table 8 materials-15-04956-t008:** Measured Rockwell hardness on the surface of the surface layers for samples prepared by each MMC powder.

Specimen	Surface Hardness		
	Mean, HV1	Mean, HRC	Standard Deviation
1-PG	462	46.3	0.5
2-PG	475	47.3	2.6
3-PG	480	47.7	2.5
4-PG	488	48.3	1.2
1-PE	660	58.3	3.7
2-PE	556	52.7	3.3
3-PE	602	55.3	2.9
4-PE	610	55.7	1.2

**Table 9 materials-15-04956-t009:** Abrasive wear resistance tests results.

Sample ID	Mass before Test, g	Mass after Test, g	Average Mass Loss, g	Material Density, g/cm^3^	Average Volume Loss, mm^3^	Relative Abrasive Wear Resistance *
**Surface layers prepared with PG powder, NiSiB+60%WC**						
1-PG	195.6418	195.3264	0.3154	11.1935	28.1771	4.7
2-PG	209.0038	208.7471	0.2567	11.1935	22.9329	5.7
3-PG	196.0594	195.6905	0.3689	11.1935	32.9566	4.0
4-PG	227.8358	227.5179	0.3179	11.1935	28.4004	4.7
**Surface layers prepared with PE powder, NiCrSiB+45%WC**						
1-PE	226.4951	226.1412	0.3539	9.8274	36.0116	3.7
2-PE	228.6697	228.3604	0.3093	9.8274	31.4732	4.2
3-PE	231.6575	230.8754	0.7821	9.8274	79.5836	1.7
4-PE	221.7090	221.3348	0.3742	9.8274	38.0772	3.5
**Reference Marterial AR400 steel**						
H1 H2	104.6219 111.7377	103.4971 110.7989	1.0318	7.7836	132.5607	1.0

* Abrasive wear resistance relative to reference material AR400.

**Table 10 materials-15-04956-t010:** Optimum process parameters used for sample preparation, applicable to the two MMC powders used for this study.

Process Parameter	Value	Unit
Plasma arc current	110	A
Pilot arc current	30	A
Travel speed	1.3	mm/s
Plasma gas flow rate	1.2	L/min
Open circuit voltage	95	V DC
Power flow (Cos phi)	0.99	-

## Data Availability

Not applicable.

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
