# Peer review of "Powder Plasma Transferred Arc Welding of Ni-Si-B+60 wt%WC and Ni-Cr-Si-B+45 wt%WC for Surface Cladding of Structural Steel"

_materials, 2022, doi:10.3390/ma15144956_

Round 1

Reviewer 1 Report

The authors have studied the powder plasma transferred arc welding of Ni-Si-B+60wt%WC and Ni-Cr-Si-B+45wt%WC for surface cladding of structural steel. This work is interesting for microjoining industries but there are some issues that need to be addressed before publication.  

1.      The highlight of Table 1 is not clear. It should be shifted to other section, preferably in Experimental.

2.      The rationale behind the selection of 60% and 45% WC should be mentioned.

3.      Figure 2 is just a showcase material. The various parts should be highlighted. Alternatively, an schematic could have been better.

4.      What is the logic behind the demonstration of Figure 3 a and b? No big difference can be noticed.

5.      Where are the carbides and matrix particles in Figure 5? How do you guess their identities? Please check the magnification 200X is same or different.

6.      x-radiation energy diagram=> It should be X.

7.      Scattered X-ray dispersion energy (EDS) => please check it.

8.      Figure 9 is too grainy. Please supply a better image with good sharpness.

9.      Authors are presenting the hardness in different units HV and HRC. Please be consistent throughput the manuscript.

10.   What is the wear mechanism of WC free sample? The wear mechanism should be elaborated.

11.   The conclusions should be bulleted. Only important conclusions should be presented for clarity.

Reviewer 2 Report

the used methods are pertinet, the obtained results and the associated discussion are very interesting. 

Some details need to be added to lead a better understanding:

- in paragraph 2.2.2, it is necessary to precise the test protocole: test zone, surface preparation, test number, the associated test precsion. For macrohardness test, it is necessary to discuss the influence of the structure gradient on the obtained results, for microhardness, it is important to discuss the results validity in considering the footprint size and the particle size;

- the paragraph 2.3.1 should be removed because any results have been presented afterwards; 

- in caption of figure 8 and ta ble 4, it is necessary to precise the studied sample and their preparation conditions; the digital values presented in table 4 should be associated with precison/error;

- in caption of table 5, it is necessary to precise the measurement zone (at extreme surface ? in corss-section ? at which layer ?)

- for all results presented in table 7, it is necessary to add the measurement precison, and discuss the physical meaning of the 4 digitial numbers after the point in considering to the announced precision;

- in caption of table 8, it is important to precise the optimum parameters are applicable for the two studied powders.

Round 2

Reviewer 1 Report

The authors have done a great job. The revised manuscript can be accepted for publication. 

Reviewer 2 Report

in the revised version of the manuscript, author has made necessary corrections incduling all remarks from reviewers.
